# Formaldehyde and Acetaldehyde Exposure in “Non-Traditional” Occupational Sectors: Bakeries and Pastry Producers

**DOI:** 10.3390/ijerph20031983

**Published:** 2023-01-21

**Authors:** Lucia Miligi, Sara Piro, Chiara Airoldi, Renato Di Rico, Raffaella Ricci, Rudy Ivan Paredes Alpaca, Fabrizio De Pasquale, Angela Veraldi, Alessandra Ranucci, Stefania Massari, Alessandro Marinaccio, Giorgia Stoppa, Anna Cenni, Cinzia Trane, Antonio Peruzzi, Maria Cristina Aprea

**Affiliations:** 1Occupational and Environmental Epidemiology Branch, Cancer Risk Factors and Lifestyle Epidemiology Unit, Institute for Cancer Research, Prevention and Clinical Network (ISPRO), 50139 Florence, Italy; 2Local Health Unit AUSL Modena, SPSAL-Department of Public Health, 41121 Modena, Italy; 3Department of Occupational and Environmental Medicine, Epidemiology and Hygiene, Italian Workers’ Compensation Authority (INAIL), 00143 Rome, Italy; 4Unit of Occupational Hygiene and Toxicology, Public Health Laboratory, Department of Prevention, AUSL South-East Tuscany, 53100 Siena, Italy

**Keywords:** formaldehyde, formaldehyde measurements, productive sectors, bakeries, pastry industry, acetaldehyde, aldehydes, chemical exposure, leavening, work processes

## Abstract

Introduction. Formaldehyde, a colorless and highly irritating substance, causes cancer of the nasopharynx and leukemia. Furthermore, it is one of the environmental mutagens to which humans are most abundantly exposed. Acetaldehyde was recently classified as carcinogen class 1B and mutagen class 2 in Annex VI EC regulation. Occupational exposure to the two aldehydes occurs in a wide variety of occupations and industries. The aim of this study is to deepen exposure to the two aldehydes in the non-traditional productive sectors of bakeries and pastry producers. Methods. The evaluation of exposure to formaldehyde and acetaldehyde was conducted in Italy in 2019, in specific tasks and positions of 11 bakeries and pastry producers (115 measures, of which 57.4% were in fixed positions and the rest were personal air sampling). The measurements were performed using Radiello© radial diffusion samplers. A logarithmic transformation of the data was performed, and the correlation between the two substances was calculated. Moreover, linear models considering the log-formaldehyde as the outcome and adjusting for log-acetaldehyde values were used. Results. The study identified high levels of acetaldehyde and formaldehyde exposure in the monitored workplaces. Higher mean values were observed in the leavening phase (8.39 µg/m^3^ and 3.39 µg/m^3^ for log-transformed data acetaldehyde and formaldehyde, respectively). The adjusted univariate analyses show statistically significant factors for formaldehyde as the presence of yeast, the presence of type 1 flour, the use of barley, the use of fats, the type of production, the use of spelt, and the presence of type 0 flour. Conclusions. The measurements confirmed the release of formaldehyde and acetaldehyde in bakeries and pastry industries, especially in some phases of the work process, such as leavening.

## 1. Introduction

Formaldehyde is one of the compounds recognized as carcinogenic to humans in workplaces and living environments. It is a ubiquitous chemical agent at “indoor” and “outdoor” levels. The International Agency for Research on Cancer (IARC) includes formaldehyde in group 1 as “carcinogenic to humans”, confirming sufficient evidence in humans of carcinogenicity for nasopharyngeal cancer and leukemia, limited evidence for nasal sinus cancer, and sufficient evidence in experimental animals [1]. The European Union (ECHA) ranks formaldehyde in category 1B (since 1 January 2016) as a carcinogen with a CLP concentration limit ≥ 0.1%. According to the harmonized classification and labeling (Adaptation to Technical Progress ATP06) approved by the European Union, this substance is toxic if swallowed, is toxic in skin contact, causes severe skin burns and eye damage, is toxic if inhaled, may cause cancer, is suspected of causing genetic defects, and may cause allergic skin reactions [2]. Formaldehyde is used mainly in the production of various types of resin. Phenolic, urea, and melamine resins have wide uses as adhesives and binders in wood furniture production, the pulp and paper industry, the synthetic vitreous fiber industry, the production of plastics and coatings, and textile finishing. Polyacetal resins are widely used in the production of plastics [1]. Formaldehyde is also used extensively as an intermediate in the manufacture of industrial chemicals. Formaldehyde is used directly in an aqueous solution (known as formalin) as a disinfectant and preservative in many applications [1]. Formaldehyde is found as a natural product in most living systems and in the environment. It is naturally present in fruits and some foods, and it is developed endogenously in humans as a consequence of oxidative metabolism. In addition, combustion processes, e.g., emissions from motor vehicles, power plants, incinerators, refineries, wood stoves, and kerosene heaters, are common non-occupational sources of exposure to formaldehyde.

As reported in the IARC monograph [1], the highest continuous exposures have been measured with the varnishing of furniture and wooden floors, textile finishing, the garment industry, the treatment of fur, and certain jobs within manufactured board mills and foundries. Short-term exposures to high levels have been reported for embalmers, pathologists, and paper workers. Lower concentrations have usually been encountered during the manufacture of man-made vitreous fibers, abrasives, and rubber and in formaldehyde-production industries. A very wide range of exposure levels has been observed in the production of resins and plastic products. In recent decades, the development of resins with a lower release of formaldehyde and the improved ventilation in workplaces resulted in lower exposure levels of workers in many industrial settings [1].

It has been estimated that, in the early 1990s, a substantial proportion of workers in the EU had been exposed to carcinogens in the past [3]; some of these causative agents are still present in workplaces, and formaldehyde is one of them.

Short-term exposure to acetaldehyde is considered mildly irritating to the eyes, skin, and respiratory tract. Inhalation can cause effects on the central nervous system and tissue lesions in the respiratory tract. Repeated or prolonged contact with the skin can cause dermatitis. The IARC classified it in class 2B as a possible human carcinogen [4]. In 2010, the IARC evaluated the risk of cancer due to alcohol consumption, including acetaldehyde, confirming that there was sufficient evidence of carcinogenicity in animal experiments [5]. Moreover, in 2012, the IARC concluded that acetaldehyde associated with alcohol consumption is carcinogenic to humans (group 1) [6]. Recently, acetaldehyde was ranked by the ECHA (ATP06) for carcinogenicity in class 1B (presumed to have carcinogen potential in humans) and mutagens class 2 (suspected of causing genetic defects) [7].

To the best of the authors’ knowledge, there are no studies on the assessment of formaldehyde in pastry or bread shops and only one study on acetaldehyde exposure to workers involved in baking processes [8]. The emission of these aldehydes in grocery stores and pastry shops is cited, respectively, in a US study [9] and a study aimed at evaluating the concentration and distribution of carbonyl compounds in certain areas of a bus terminal in Brazil [10]. On the other hand, acetaldehyde is widely described as one of the volatile components of the aroma of bread. The type of flour, cereals used, processing phases, and cooking treatment have significant effects on the generation of flavoring substances [11].

Table 1 presents a short description of the key elements that characterize the studies in which aldehyde assessment was performed in baking processes.

Recently, in the Tuscany region (central Italy), during the investigation of a case of nasopharyngeal cancer in a subject who had always worked in a bakery, an unknown exposure was discovered: the release of formaldehyde and acetaldehyde during the fermentation and baking of bread. This exposure was detected in a preliminary study not only in the bakery where the subject was employed but also in other workplaces where measurements of formaldehyde concentration in the air were conducted at personal and environmental levels [12]. Following these unexpected results and the possible association between nasopharyngeal cancer and exposure to formaldehyde, we decided to deepen the knowledge of exposure to formaldehyde and acetaldehyde in the “non-traditional” occupational sector of bakeries and pastry workplaces. The aim of the study presented here was to extend the measurements of the two aldehydes in other workplaces characterized by baking processes. 

## 2. Materials and Methods

The 115 samples analyzed were collected between February and October 2019 in 11 bakeries and pastry cook companies located in two Central Italy regions: Tuscany and Emilia Romagna. At the same time as the environmental samplings, information on the companies was collected using a questionnaire that was previously structured. The principal items of the questionnaire were the type of company (artisan or industrial), the number of workers, the number of premises, the type of oven, the presence of thermoplastic packaging machines, ventilation, leavened process rooms, the quantity and type of flours, bread recipe, and a short description of production.

In each company, the samplings were performed in a single day. Personal and environmental samplings were performed using a radial diffusive sampler Radiello© for aldehydes (florisil cartridge containing the derivatizing reagent 2,4 dinitrophenylhydrazine), placed on the outer garment collar and in the center of the workplace or near specific emission sources, respectively. The samplings lasted the whole operation or specific task; only in a few cases, short-term samples (15 min) were collected in a fixed position in leavening cells or as personal sampling.

The analysis, after extraction of the Radiello© cartridge with acetonitrile, was performed in liquid chromatography with a reversed-phase column and photodiode detector, using an analytical method accredited by ACCREDIA (the only Italian accreditation body for test laboratories). The analytical uncertainty, expressed as extended uncertainty at 95% probability (coverage factor 2), was around 10% throughout the full measurement range.

For each qualitative variable recorded in the questionnaire and analyzed, we reported absolute and relative frequencies, while for numerical variables, means and standard deviation or median and interquartile ranges were reported. We performed a logarithmic transformation of acetaldehyde and formaldehyde air concentration in order to obtain a normal distribution; the ratio between the two measures (formaldehyde/acetaldehyde) was also calculated. Moreover, the Pearson index was estimated to evaluate the presence of a correlation between the two substances. 

To better show the distribution of log-acetaldehyde and log-formaldehyde concentrations between different production phases, box plots were also drawn. Finally, a univariate analysis was performed, considering the log-formaldehyde as the outcome and evaluating the association among categorical variables adjusted for log-acetaldehyde values. Variables with *p*-values < 0.05 were included in a multivariate model, and then we considered the only one that retained the statistical significance. Beta and 95% confidence intervals (95% CI) were reported for each parameter. 

In addition, *p*-values less than 0.05 were considered statistically significant, and two side tests were performed; all analysis was conducted using the software SAS 9.4.

To synthesize the methods used, a workflow chart is presented in Figure 1.

## 3. Results 

A total of 115 samples were collected from 11 companies of bakers and pastry cook industries. In the Emilia Romagna region, the province involved was Modena (eight companies, one with two different productions); in the Tuscany region, the provinces involved were Siena (one company), Grosseto (one company), and Arezzo (one company). Five industries (45.4%) were pastry shops; three were bakeries (27.3%); two (18.2%) produced “tigelle” and “piadine”, special Italian bread; and one (9.1%) prepared pizza. For each company, a minimum of 4 and a maximum of 17 samples were recorded. In Table 2, the principal characteristics of the sampled firms are described.

Of the 115 samples, 66 (57.4%) were environmental, while the remaining (n = 49, 42.6%) were personal. Samplings were performed in different phases: leavening (n = 34, 29.6%); shaping, dividing, and packaging (n = 30, 26.1%); dough (n = 21, 18.3%); blank in the sales room (n = 8, 7%); baking (n = 6, 5.2%); shift manager and jolly worker (n = 6, 5.2%); and other (n = 10, 8.7%). 

A detailed focus was obtained on formaldehyde and acetaldehyde: the concentration of formaldehyde ranged from 6.18 to 475 µg/m^3^, while acetaldehyde ranged from 8.26 to 21,841 µg/m^3^, indicating a wide dispersion of values. The concentrations (mean ± SD) of log-formaldehyde and log-acetaldehyde were 3.12 ± 0.81 µg/m^3^ and 7.31 ± 1.65 µg/m^3^. The mean ratio of acetaldehyde to formaldehyde was 149 ± 209, and the median was 64.8 (IQR 31.19; 171.41), while the mean ratio of log-transformed variables was 2.45 ± 0.66, indicating that more of the former was present. Moreover, the two log-transformed variables were positively correlated with a Pearson correlation value of 0.45 (*p* < 0.001). No significant correlation was found between the two aldehydes and the average room temperature (*p* = 0.201 and *p* = 0.888 for log-formaldehyde and log-acetaldehyde, respectively).

Table 3 shows the descriptive statistics obtained for the log-acetaldehyde and log-formaldehyde concentrations in the different categories. Higher values were observed for some of the different phases; in particular, the leaving phase shows higher acetaldehyde and formaldehyde concentrations. Other factors increasing aldehyde concentration were the use of brewer’s yeast, the artisanal type of firm, and the use of fat (margarine). The production of pizza showed the highest acetaldehyde concentration, and the production of “tigelle” had the highest formaldehyde concentration.

Considering the productive phases, higher values of log-aldehydes, in terms of means, were observed in the leavening phase (8.39 µg/m^3^ and 3.39 µg/m^3^ for acetaldehyde and formaldehyde, respectively), while lower values were in the blank sales counter zone, considered the non-exposed area (4.65 µg/m^3^ and 2.67 µg/m^3^ for acetaldehyde and formaldehyde, respectively). A visual representation is reported in Figure 2.

The companies with high acetaldehyde/formaldehyde ratios were those in which high-leavening processes were performed for the production of croissants or a second fermentation was necessary, as in the case of pizza production.

After a first description of the two substances, adjusting for log-acetaldehyde, univariate analyses were performed, considering log-formaldehyde as the outcome and including the covariates. The presence of yeast, the presence of type 1 flour, the use of barley, the use of fats, the type of production (*p*-values less than 0.001), the use of spelt (*p* = 0.005), and the presence of type 0 flour (*p* = 0.030) were significantly associated with the outcome. In a multivariable model, the only variables maintained were the type of production and type of yeast with an R^2^ of 0.43 (see Table 4).

## 4. Discussion

Aldehydes are highly reactive electrophile compounds present in air and in products intended for use and consumption. In general, aldehydes exposure occurs outdoors and indoors, including workplaces. Formaldehyde has often been indicated as the main cause of “poor or bad indoor air quality” and the focus of researchers for years dealing with pollution in confined spaces. Pollution is generally detected in higher concentrations inside buildings than in the external environment, where concentrations vary significantly depending on the geographical area considered. Against a background value in rural environments of about 1 µg/m^3^, practically the entire population is exposed to much higher indoor levels, with a median of 26 ± 6 µg/m^3^; it is estimated that at least 20% of the European population exceeds the NOAEL (no-observed-effect-level) of 30 µg/m^3^ [13].

For living environments, WHO recommends a value of 100 µg/m^3^ for an average exposure of over 30 min to prevent sensory irritation in the general population. This value is also able to prevent effects on lung function and long-term effects [14].

Regarding occupational exposure to formaldehyde, Directive (EU) 2019/983 of the European Parliament and of the Council of 5 June 2019, amending Directive 2004/37/EC on the protection of workers from the risks related to exposure to carcinogens or mutagens at work, established an 8 h occupational limit of 370 µg/m^3^ and a 15 min limit of 740 µg/m^3^. The ACGIH reports a TLV-TWA of 123 µg/m^3^ and a TLV-STEL of 370 µg/m^3^ [15].

For acetaldehyde, the median concentration of 10–20 µg/m^3^, found indoors in the available European studies, is of the same order of magnitude as the levels observed in the exhaled air of the general population as a result of its endogenous production, without considering increases due to the consumption of alcoholic beverages and tobacco smoking [13].

The ACGIH reports a ceiling limit of 45,000 µg/m^3^.

We can hypothesize that the origin of formaldehyde and acetaldehyde during the production of baked goods is linked to the complex chemical reactions that occur during the leavening and cooking phases. The fundamental transformation that takes place during leavening (alcoholic fermentation) leads to the production of ethyl alcohol and carbon dioxide. At the same time, however, various additional products are also formed, some of which originate from sugars and others come from amino acids. The amino acid, because of a process of hydrolysis, undergoes a deamination releasing ammonia and the resulting hydroxy acid, which splits into formic acid and an aldehyde with one less carbon atom than the starting amino acid [16].

The baking phase of the bread and the formation of the crust also involve known chemical reactions. The Maillard reaction is a complex chemical process that involves proteins and sugars as a result of the action of high temperatures. In summary, the carbonyl group of reducing sugars reacts with the amino group of proteins, forming an unstable condensation product (Schiff base) from which rearrangement and decomposition processes start, leading to the formation of carbonyl compounds, among others [17].

The data obtained for formaldehyde and acetaldehyde in the various processing stages of bakery products are clearly higher than those usually found indoors in European studies; they are also significantly higher than those observed in grocery stores (average values of 165 µg/m^3^ for acetaldehyde and 26 µg/m^3^ for formaldehyde) [9] and in pastry shops (51.8 ± 39.7 µg/m^3^ for acetaldehyde and 30.6 ± 13.7 µg/m^3^ for formaldehyde), for which the authors report as a source of acetaldehyde production the frying oils and the leavening/cooking of bakery products [10].

Also, in a Chinese bakery study [8], high levels of acetaldehyde concentrations were found; the authors of this study reported that the acetaldehyde measured was 37–83 times higher than what was reported from other studies for residential areas and Chinese restaurants.

The maximum formaldehyde values in the leavening area exceeded the 100 µg/m^3^ recommended by the WHO for living environments, the TLV-TWA of 123 µg/m^3^ reported by ACGIH, and the 8 h occupational limit of 370 µg/m^3^ established by Directive (EU) 2019/983.

The acetaldehyde/formaldehyde ratio is, on average, higher than 20 in the processing stages, confirming that the main carbonyl compound formed is acetaldehyde, but formaldehyde cannot be neglected as one of the products of the chemical reactions in leavening and cooking.

The data obtained allow us to generically state that the leavening phase caused the maximum release of acetaldehyde and formaldehyde. It is equally true that the 11 companies evaluated showed airborne concentrations of the two aldehydes very different from each other, with apparently inexplicable peaks of formaldehyde, especially in the production of “tigelle” and “crescentine” but also of pizza and other products that require high leavening.

Many factors can influence the concentration levels of the two carbonyl compounds. Among these, we can generally include ventilation, the volume of the production rooms, the separation between the leavening phases, and the other phases of the production process. Artisanal firms with smaller rooms and all the production phases carried out in the same room show higher levels of formaldehyde compared to industrial production. Floor area and ventilation system are factors already considered in previous studies [8]. Regarding the recipes of baked products, it is possible to hypothesize that the leavening of doughs containing large quantities of animal fats causes a greater release of formaldehyde. It is not possible, with the available data, to establish whether the type of flour used affects the release of formaldehyde, as in most cases, various types of flour are used. The same consideration applies to the type of yeast. Regarding the temperature of the leavening cell, it seems possible to hypothesize that slow leavening at low temperatures causes a greater production of formaldehyde. Furthermore, the maturation of the product (second leavening) seems to have a certain influence on the release of this pollutant.

The average and maximum levels observed in this study for formaldehyde exposure are, however, significantly lower than those in other work sectors, such as foundry workers [18], plastic laminate plant workers [19], employees of a wood industry [20], and pathology departments of hospitals [21].

The concentrations of formaldehyde detected in this study are instead significantly higher than those observed during the thermo-decomposition of plastics [22].

With regard to acetaldehyde, the exposure found in workers in food manufacturing industries is comparable with that observed in this study [23].

## 5. Conclusions

Formaldehyde is a ubiquitous environmental chemical carcinogen with a high number of exposed workers worldwide. Among the approximately 32 million workers (23% of the total employed) in the 15 countries of the EU exposed to the agents covered by CAREX in 1990–3, 990,000 workers were estimated to be exposed to formaldehyde [3] and about 113,000 in Italy [24]. The national estimates of exposure to formaldehyde in Italian workplaces, provided by the Italian Workers’ Compensation Authority (INAIL) and based on a total of 1610 measurements selected from the Italian database on occupational exposure to carcinogens in the period 1996–2014, show that occupational exposure to formaldehyde occurs in a variety of different sectors, but workers currently at higher risk are those employed in the health care sector and wood processing industry [25].

The deepening of a baker’s nasopharyngeal cancer case found in Tuscany, Italy [12], and the measurements taken in bakeries provided new information on the presence of formaldehyde and acetaldehyde in unknown productive sectors such as bread or cake production.

The present study of the bakery and pastry industries considered two Italian regions, Tuscany and Emilia Romagna, and the evaluation of different production (bread, cakes, and other products) provided new information on the exposure to aldehydes in this productive sector. The presence of the two aldehydes was found in different bread and cake productions, with higher values of carcinogenic agents observed into the leaving phase.

To better understand this exposure in the context of bread and cake production, further investigations are needed on the importance of all the variables that can influence the levels of exposure of formaldehyde and acetaldehyde in these workplaces, such as the use of different types of flours and fats, the presence or absence of ventilation, the size of the premises, etc.

Furthermore, the contextual exposure to elevated concentrations of acetaldehyde, as demonstrated by the results, may have some pathogenetic significance and will have to be investigated [26,27].

The exposure to two carcinogenic agents in this productive sector is a valid occupational health and safety concern, and proper actions should be taken to protect workers.

The data collected indicate the need to inform workers, reduce the pollution of the workplace during the production of bread, e.g., by delimiting the leavening phase, and achieve a good exchange of ambient air in order to reduce the stagnation of pollutants and the possible effects on health.

## Figures and Tables

**Figure 1 ijerph-20-01983-f001:**
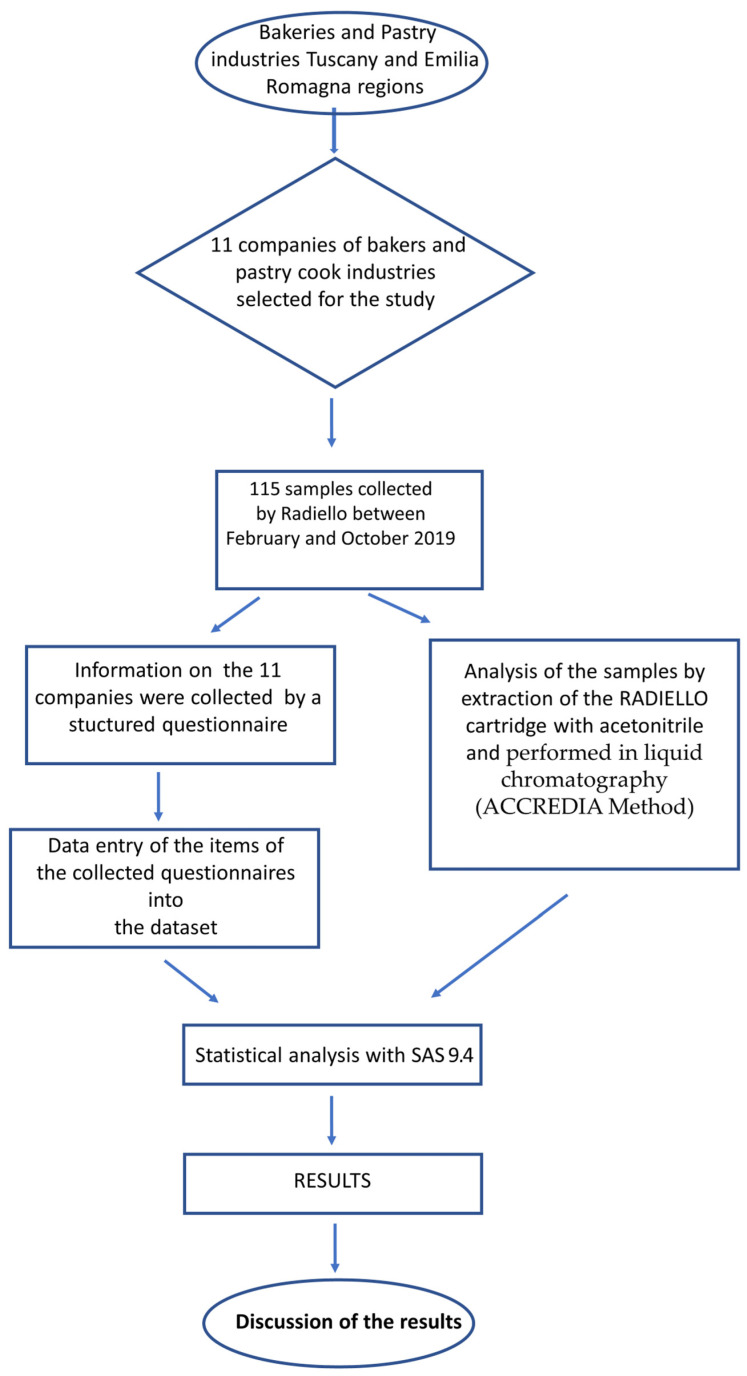
Workflow chart.

**Figure 2 ijerph-20-01983-f002:**
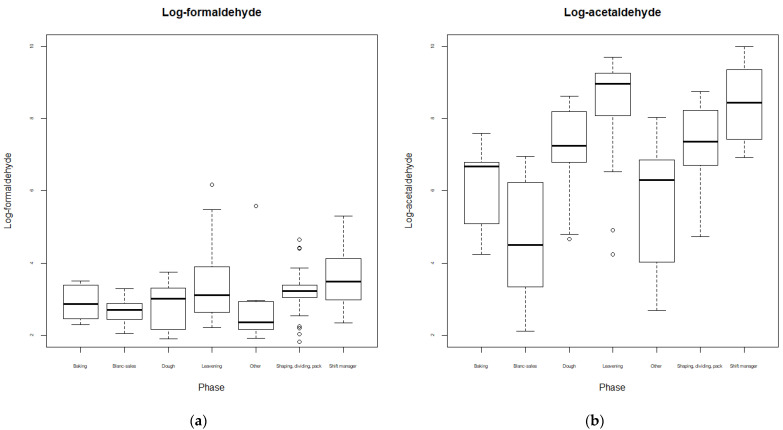
The boxplots show the level of log-formaldehyde by the phase of the production cycle (**a**) and log-acetaldehyde by the phase of the production cycle (**b**).

**Table 1 ijerph-20-01983-t001:** Description of studies in which aldehydes assessment was performed in baking processes.

Author and Number of Reference	Year	Country	Objective	Analytical Method	Statistical Method	Factors Considered	Results
**Chang P.T [8]**	2018	Taiwan	To determine the airborne pollutants in five bakeries in Taiwan	Air sampling methods were used according to NIOSH and OSHA methods. Three different sites (weighing area, baking area, dough making/decorating office/counter area) were taken for air sampling.	The mean and standard deviation of the concentration measured were calculated; Kruskal-Wallis tests and Dunn’s multiple comparisons were also performed to compare the air sampling and analysis data from different sampling sites.	Floor area, ventilation system, personal protective equipment used, types of procedures of baking processes, number of workers per day ingredients	Number of samples 5/20 Mean concentration (mg/m^3^) ± SD 0.19 +/− 0.26 The concentrations of acetaldehyde measured in these bakeries were 37–83 times higher than what was reported from other studies.
**Nirlo E.L. [9]**	2014	US: Pennsylvania and Texas	To investigates the variability in VOC concentrations in 14 individual stores over the course of a year	Formaldehyde and acetaldehyde were collected and analyzed using the methods and procedures described US EPA compendium method TO-11 A.	Formaldehyde WBER (whole-building emission rate) was calculated in accordance with the equation using concentrations and air exchange rates measured during the sampling event.	Formaldehyde and acetaldehyde indoor	High levels of acetaldehyde in two of the mid-sized grocery stores with a concentration of 92 ppb, possibly due to the preparation of dough and baking activities. Formaldehyde is the most important contaminant of concern in retail stores. In this study, 14 stores exceeded the most conservative health guideline for formaldehyde. The vast majority of VOCs were present in retail stores at low concentrations, but episodic activities, such as cooking and cleaning, could lead to relatively high indoor concentrations of ethanol, acetaldehyde, and terpenoids. Based on experiments in two stores, it appeared that increased ventilation could reduce VOC concentrations in retail stores, although the concentration decrease might not be proportional to the ventilation increase for some compounds.
**de Mendonca O.S. [10]**	2014	Brazil	To describe the determination of 30 carbonyl compound (CC) in three areas (bus terminal, passengers circulation area, and a pastry shop) and in an open area within 700 m of the terminal	Samples were collected using SEP PAK cartridges. Air sampling followed the EPA TO 11 A protocol.	A statistical package was used to evaluate the distribution of CC in the study sites. The resulting model was in terms of variations around the means and is often a preferred preprocessing method because it focuses on differences between observations rather than absolute values.		The largest mean concentration of CC in the bus terminal was found in the pastry shop (159 +/− 56.1 µg/m^3^); the high concentration of acetaldehyde found here might be a result of its emission from cooking oils used for frying and as a by-product of baking yeast fermentation. Acetaldehyde predominated in the pastry shop located in the terminal, possibly owing to the cooking activities (baking and frying) carried out here. In the pastry shop, formaldehyde concentration was 30.6 +/− 13.7, and acetaldehyde was 51.8 +/− 39.7 µg/m^3^.

**Table 2 ijerph-20-01983-t002:** Principal characteristics of the sampled companies.

Description of the Sampled Firms					
Company	Type of Company	Principal Production	Location (City)	Region	Number of Samples
n. 1	Industrial	Piadine and tigelle	Modena	Emilia Romagna	9
n. 2	Industrial	Bakery and pastry products	Arezzo	Tuscany	14
n. 3	Small industry	Tigelle, organic and produced with conventional flour	Modena	Emilia Romagna	11
n. 4	Industrial	Pre-cooked bread	Modena	Emilia Romagna	8
n. 5	Artisanal	Pastry	Siena	Tuscany	7
n. 6	Industrial	Rusks, croissants, *panettone,* and *colomba* (typical Italian cakes for Christmas and Easter)	Grosseto	Tuscany	13
n. 7	Industrial	Sandwiches	Modena	Emilia Romagna	10
n. 8	Artisanal	Brioches	Modena	Emilia Romagna	4
n. 9	Artisanal	Brioches and *cannoli* (a special Italian cake)	Modena	Emilia Romagna	7
n. 10	Industrial	Pizza	Modena	Emilia Romagna	15
n. 11	Artisanal	Brioches	Modena	Emilia Romagna	17

**Table 3 ijerph-20-01983-t003:** Mean, standard deviation, and *p*-values for acetaldehyde and formaldehyde concentration by the log-transformed variables.

Variable	Level	N (%)	Log-Acetaldehyde	Log-Formaldehyde
			Mean ± SD Log (µg/m^3^)	Mean ± SD Log (µg/m^3^)
		N = 115		
Measurements	Environmental sampling	66 (57.39)	7.21 ± 1.95	3.11 ± 0.810
	Personal sampling	49 (42.61)	7.43 ± 1.15	3.13 ± 0.830
*p*-value			0.4970	0.9099
Productive phases	Blank sales counter	8 (6.96)	4.65 ± 1.76	2.67 ± 0.390
	Shaping, dividing, packaging	30 (26.09)	7.39 ± 0.980	3.21 ± 0.650
	Baking	6 (5.22)	6.17 ± 1.25	2.9 ± 0.550
	Sough	21 (18.26)	7.21 ± 1.15	2.83 ± 0.610
	Shift manager and jolly worker	6 (5.22)	8.43 ± 1.15	3.63 ± 1.05
	Leavening (fermentation)	34 (29.57)	8.39 ± 1.32	3.39 ± 0.940
	Other phases	10 (8.70)	5.69 ± 1.68	2.80 ± 1.18
*p*-value			<0.001	0.0445
Use of flour				
Type 0	No	53 (46.09)	7.81 ± 1.34	3.38 ± 0.610
	Yes	62 (53.91)	6.87 ± 1.78	2.88 ± 0.910
*p*-value			0.0020	0.0011
Type 00	No	40 (34.78)	6.76 ± 1.55	2.98 ± 0.970
	Yes	75 (65.22)	7.60 ± 1.64	3.2 ± 0.710
*p*-value			0.0093	0.1749
Fats	Margarine, egg	48 (41.74)	7.84 ± 1.34	3.15 ± 0.810
	Oil	47 (40.87)	7.2 ± 1.66	2.99 ± 0.580
	Lard	9 (7.83)	5.58 ± 2.27	3.53 ± 1.67
	Lard, oil	11 (9.57)	6.82 ± 1.21	3.16 ± 0.480
*p*-value			0.0008	0.3360
Yeast	Brewer’s yeast	77 (66.96)	7.6 ± 1.67	3.36 ± 0.780
	Brewer’s yeast / sourdough	21 (18.26)	6.38 ± 1.69	2.46 ± 0.440
	Sourdough/natural	4 (3.48)	7.05 ± 0.690	3.67 ± 0.280
	Natural	13 (11.30)	7.12 ± 1.14	2.29 ± 0.320
*p*-value			0.0236	<0.001
Semola	No	86 (74.78)	7.55 ± 1.63	3.22 ± 0.840
	Yes	29 (25.22)	6.58 ± 1.53	2.76 ± 0.580
*p*-value			0.0060	0.0156
Type of production	Bakeries	25 (26.60)	7.08 ± 1.27	3.11 ± 0.360
	Pastry production	41 (43.62)	7.90 ± 1.41	3.12 ± 0.870
	Production of pizza	15 (18.96)	8.07 ± 1.57	3.20 ± 0.520
	“Tigelle” and “piadine” production	13 (13.83)	6.03 ± 2.01	3.58 ± 1.37
*p*-value			0.0005	0.3346
Type of firm	Artisan	35 (30.43)	8.11 ± 1.33	3.46 ± 0.700
	Small industry	11 (9.57)	6.82 ± 1.21	3.16 ± 0.480
	Industry	69 (60.00)	6.97 ± 1.73	2.92 ± 0.860
*p*-value			0.0020	0.0059
Ventilation	Mechanical	71 (61.74)	7.26 ± 1.64	3.11 ± 0.850
	Natural	44 (38.26)	7.38 ± 1.69	3.14 ± 0.740
*p*-value			0.7130	0.8546

**Table 4 ijerph-20-01983-t004:** Adjusted (log-acetaldehyde) multivariable model considering log-formaldehyde as outcome. Beta and 95% confidence interval were reported.

Variable		Beta [95% CI]	*p*-Value
Log-acetaldehyde		0.24 [0.15; 0.33]	<0.0001
Yeast	Brewer’s yeast vs. natural	0.95 [0.51; 1.39]	0.0005
	Brewer’s yeast / sourdough vs. natural	0.73 [0.02; 1.45]	
	Sourdough/natural vs. natural	0.74 [−0.17; 1.65]	
Type of production	Bakeries vs. “tigelle”	−0.72 [−1.26; −0.18]	0.0188
	pastry production vs. “tigelle”	−0.67 [−1.21; −0.12]	
	production of pizza vs. “tigelle”	−0.93 [−1.52; −0.35]	

## Data Availability

The data presented in this study are available on request from the corresponding author.

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
