# Peer review of "Formaldehyde and Acetaldehyde Exposure in “Non-Traditional” Occupational Sectors: Bakeries and Pastry Producers"

_ijerph, 2023, doi:10.3390/ijerph20031983_

Round 1
Reviewer 1 Report
This manuscript entitled ‘Formaldehyde and acetaldehyde exposure in “non-traditional” occupational sectors: bakeries and pastry producers’ focused on exposure of formaldehyde and rare acetaldehyde in grocery stores and pastry shops. This is an interesting investigation. However, the data were showed with exceeding simple and without confidential statistical analysis.
Major concerning:
1. Why did the authors focus the airborne pollution in the special occupational sectors? Please add this in the background.
2. What is the difference of formaldehyde and acetaldehyde in different phase based on the possibility values in figure 1 and figure 2.
3. The possibility values about the difference analysis showed be added in table 1.
4. What the possible sources for formaldehyde and acetaldehyde existence in such places, are there special relation to the bakery production methods?
5. The whole text should be improved with carefully revision based on good logistic, such as distribution map of sample sites.
Author Response
RESPONSE TO REVIEWER 1
This manuscript entitled ‘Formaldehyde and Acetaldehyde exposure in “non-traditional” occupational sectors: bakeries and pastry producers’ focused on exposure of formaldehyde and rare Acetaldehyde in grocery stores and pastry shops. This is an interesting investigation. However, the data were showed with exceeding simple and without confidential statistical analysis.
Major concerning:
- Why did the authors focus the airborne pollution in the special occupational sectors? Please add this in the background.
A: In response to the reviewer’s comment, we explained more in deep in the introduction section our interest in investigating the exposure to aldehydes in bakeries.
- What is the difference of formaldehyde and acetaldehyde in different phase based on the possibility values in figure 1 and figure 2.
A: In response to the reviewer's comment, we decided to modify figure 1 and figure 2 to make them comparable. Particularly we joined the figures in one panel so it’s possible compare the values of log-formaldehyde and acetaldehyde. We observed that the values of log- acetaldehyde were more variables than log- Formaldehyde.
- The possibility values about the difference analysis showed be added in table 1.
A: In response to the reviewer's comment, we deleted table 1 to avoid confusion and we reported more results into the text.
- What the possible sources for formaldehyde and acetaldehyde existence in such places, are there special relation to the bakery production methods?
A: As discussed in the conclusion section, we can hypothesize that the origin of formaldehyde and acetaldehyde during the production of baked goods is linked to the complex chemical reactions that occur during the leavening and cooking phases. The baking phase of the bread and the formation of the crust also involve known chemical reactions. The Maillard reaction is a complex chemical process that involves proteins and sugars as a result of the action of high temperatures.
- The whole text should be improved with carefully revision based on good logistic, such as distribution map of sample sites.
A: We added in the text a table with more information on the firms involved in the study. As reported in the text, the companies, object of the samplings, are located in two Regions of Central Italy, Tuscany and Emilia Romagna and their production characteristics are clearly described in table 2. The territorial dislocation does not influence the exposure of the workers; therefore, we have no considered appropriate to include a map of the sampling sites.
Reviewer 2 Report
In this study, the authors considered exposure to two aldehydes in an unusually productive industry, such as bakeries and pastry manufacturers. The analyzes of the study were made with the Radiello© radial diffusion samplers. A logarithmic transformation of the data was created. Correlation values between the two items were calculated. Finally, linear models are used. The authors argued that they confirmed the release of formaldehyde and acetaldehyde in the bakery and pastry industries at some stages of the business process, especially leavening.
The authors have extensively covered this study, but to reach the required quality, the authors should consider some of the issues I have mentioned below.
Some information that reveals the difference between this study and other studies in terms of method, purpose, and factors can be discussed in the introduction part of the study in the form of a table.
The method, factors, and analyses used can be discussed in the methodology part of the study by creating a method flow chart such as a workflow chart.
Statistical significance analyses such as the p-value showing the effect of the independent variables in the study on the dependent variable should be performed.
Statistical results such as R2, R2(ADJUSTED), and R2(PREDICTED) should be shared in the study to verify the validity of the statistical analyses made in the study.
Line:256-258: In the discussion part of the study, many factors affect the concentration levels of two carbonyl compounds, and it has been determined that the most important ones for formaldehyde are ventilation, the volume of the production rooms and the fermentation stages, and the production process. According to which statistical result was this interpretation made?
Some grammatical/typing errors should be corrected.
Author Response
RESPONSE TO REVIEWER 2
In this study, the authors considered exposure to two aldehydes in an unusually productive industry, such as bakeries and pastry manufacturers. The analyzes of the study were made with the Radiello© radial diffusion samplers. A logarithmic transformation of the data was created. Correlation values between the two items were calculated. Finally, linear models are used. The authors argued that they confirmed the release of formaldehyde and acetaldehyde in the bakery and pastry industries at some stages of the business process, especially leavening.
The authors have extensively covered this study, but to reach the required quality, the authors should consider some of the issues I have mentioned below.
Some information that reveals the difference between this study and other studies in terms of method, purpose, and factors can be discussed in the introduction part of the study in the form of a table.
A: We added in the introduction section a table with the key factors of the previous studies on aldehydes exposures in bakeries.
The method, factors, and analyses used can be discussed in the methodology part of the study by creating a method flow chart such as a workflow chart.
A: A Workflow chart was created and added in the methods section.
Statistical significance analyses such as the p-value showing the effect of the independent variables in the study on the dependent variable should be performed.
A: We thank the reviewer for the suggestion, we decided to add the rows that report the p-values.
Statistical results such as R2, R2(ADJUSTED), and R2(PREDICTED) should be shared in the study to verify the validity of the statistical analyses made in the study.
A: We implemented a multivariable model and we reported the R^2 of final model as required. Moreover, we reported the beta estimates and the p-values of the model.
Line:256-258: In the discussion part of the study, many factors affect the concentration levels of two carbonyl compounds, and it has been determined that the most important ones for formaldehyde are ventilation, the volume of the production rooms and the fermentation stages, and the production process. According to which statistical result was this interpretation made?
A: The sentence has been changed as follows: “Many factors could influence the concentration levels of the two carbonyl compounds and among these we can generally include ventilation, the volume of the production rooms and the separation between the leavening phases and the other phases of the production process: artisanal firms, with smaller rooms and all the production phases carried out in the same room show higher levels of formaldehyde compared to industrial production. Floor area and ventilation system are factors already considered in previous studies [8]”. We are sorry not to be sufficiently clear and exhaustive.
Some grammatical/typing errors should be corrected.
A: We hope that all errors have been corrected
Round 2
Reviewer 2 Report
The authors have followed most of the indications received, significantly improving the level of the paper. Well done.